# Agricultural Location and Crop Choices in China: A Revisitation on Von Thünen Model

**Hongyun Han** *[ID], **Zhen Yuan and Kai Zou**

School of Public Affairs, China Academy for Rural Development, Zhejiang University, Hangzhou 310027, China
* Correspondence: hongyunhan@zju.edu.cn; Tel.: +86-0571-88981481

**Abstract:** Growing populations and rapid urbanization have put tremendous pressure on the food supply. The rural hinterland around cities is an important source of the urban food supply chain. Facing the constraints of China's land stock, reasonable use of land space and optimization of agricultural crop structure is crucial to meet the food demand. Von Thünen Model, which is fronted by a 19th-century German economist, outlines a rural landscape of commercial farmers growing agricultural products for local markets while proposing basic patterns and principles of land use in agriculture. Using data from China's OVOP ("One village, One product"), this paper analyzes the agricultural location and crop choices around two levels of cities (provincial capital cities, and county-level cities) by using Thünen's theory. The results showed that crop density did decrease as the distance to urban increased. Crop rings are present in the vicinity of both metropolitan and county-level cities, distributed according to crop intensity. Evidence from China suggests that agricultural location and crop selection still follow the basic principles of the Thünen model. Planners and policymakers should refocus on the Von Thünen model to utilize land space and optimize agricultural production scientifically and efficiently.

**Keywords:** agricultural location; crop choices; Von Thünen Model; OVOP; one village, one product





## 1. Introduction

Von Thünen Model, as a predictive model of how rural hinterlands organize agricultural production around an urban centre, has always been a concern for geographers and agricultural economists. Though some geographers complain that the model is too simple by ignoring history and assuming a featureless plain by concentrating on the effects of transportation costs on the location of agricultural production [1]. The clearest evidence of von Thünen's model can be found today in less developed countries. Expanding populations and increasing population densities have put tremendous stress on the agricultural sector of the economies in these countries. The choice of the right crop(s) for the right type of land is the key issue that maximizes the overall contribution to meeting the food demand facing constraints of land and capital availability [2].

Broadly speaking, there are two perspectives on the Von Thünen Model of agricultural land use based on previous studies. One view believes that distance to the market plays a role in crop choices in many places [3]. The crop location is proposed as a significant factor for land value of use and intensity [4]. The Von Thünen-like pattern of concentric zones around the city is proved by Horvath [5] in Ethiopia, by Blaikie [6] in northern India, by Muller [7] in America, and by Ewald [8] in the Indian and Spanish economies. Another opposite view believes that "the Isolated State is so tortuously constructed that its central message is difficult to discern" [9] (p. 615). Abstract models could never hope to explain real-world phenomena [10]. Critics argue that the Von Thünen model is a static model unable to deal with dynamic phenomena by ignoring land ownership [10] and does not explicitly account for functional relationships between multiple urban and rural interior areas [11,12], and it is highly localized in its object of analysis [13–15]. Although "the

Thünen model and kindred recent work labelled the "new economic geography" supplies a plausible paradigm for rural economic development policy" [16] (p. 230), it is believed that Von Thünen's model has no validity in the Internet age with a globalized marketplace [10]. Associated with the change of evolving economic and resource circumstances, as improved transportation conditions have led to a decrease in transportation costs, whether the Thünen model still has explanatory power is a concern of recent research. If so, does the Thünen model still apply, and if so, to what extent it is applied to the explanation of the distribution of crop rings? Further research on the applicability of the Thünen model is needed for a better understanding of agricultural patterns in developing countries, which helps to optimize their land use and agricultural productivity [3].

Agriculture in and around urban areas is influenced by urban development [17]. In the process of China's rapidly growing urbanization, it is worth our attention to examine whether the Thünen model is still applicable, which will provide valuable highlights on land utilization. This paper provides a unique perspective on agricultural location and crop selection in the context of urbanization, using the example of "One village, One product" (OVOP). As a regional revitalization policy proposed by Governor Hiramatsu Morihiko in the Oita prefecture in Japan, the OVOP means that a village only produces one or a few kinds of agricultural products, which is termed the specialization of firms by Smith [18]. China, the first developing country to learn from Japan, has incorporated OVOP into its national agricultural policy since 2006. By 2020, China already has more than 26,000 specialized villages and towns where farmers produce the same agricultural products and are linked together to access markets through companies and cooperatives. By examining the logic of geographic location and crop choices in the development of specialized villages and towns in China under the OVOP policy, the contributions of this paper are as follows: (1) The location and crops selection of China's specialized villages and towns in different locations and its reason are firstly discussed; (2) The Thünen model is innovatively used to explain the distribution of crop varieties in China's specialized villages and towns, and crop rings centred on the metropolis and county-level cities are depicted; (3) Compared with the previous studies that only focus on the crop ring of a single city, the cross impacts from different levels of cities are distinguished and studied.

Employing an economic geography approach, this paper explores the patterns of agriculture and its role in the long-lasting regional division. The paper is organized as follows: Firstly, Thünen's models are systematically introduced and reviewed. Secondly, this article will describe the development of OVOP in China and the crop varieties of China's specialized villages and towns, followed by an analysis of crop varieties and intensity based on the Von Thünen Model. The subsequent sections are regional heterogeneity analysis and cross-sectional analysis. It ends with a brief conclusion.

## 2. Von Thünen Model

It is generally acknowledged that the roots of agricultural location theory can be traced back to 1826 when Johann Heinrich Von Thünen published his classic work, The isolated state. The ideal site consisted of completely rational (optimizing) economic behaviour, an isolated state, a single central city, settlement in villages away from the central city and a racially homogeneous population, uniform topography, uniform climate, and soil fertility, and relatively uniform and primitive transportation [19].

> "Thünen was a loner with a one-track mind. By 1803-still a minor! he had already glimpsed the equilibrium of his Isolated State: a town surrounded by a homogeneous plain, trading city goods for the rural fruits of labour and land; and with the inner rings nearest the town specializing in the goods dearer to transport, while the farther out low-rent-generating acres are growing the goods cheaper to transport" [20] (p. 1468).

Holding all other factors constant, Thünen set out to examine the influence of transport costs on the location of crop production. Taking into consideration of transportation costs, crops are concentrically arranged around the market. Summarizing the results, Von Thünen stated: "with increasing distance from The Town, the land will progressively be given up to

products cheap to transport in relation to their value" [21]. The key contribution of the Von Thünen model is that he draws attention to locational attributes of the property as a source of land rent [3]. The idea behind the Von Thünen model is very simple: various locations have different accessibility characteristics, some pieces of land are close to the market centre, and so they have an implied lower cost of transportation than the less accessible lots. As shown in Figure 1, the owner of the best land can charge more rent for a property that has lower transportation costs because those lower costs make the land very desirable and more profitable than the land away from the centre [3].

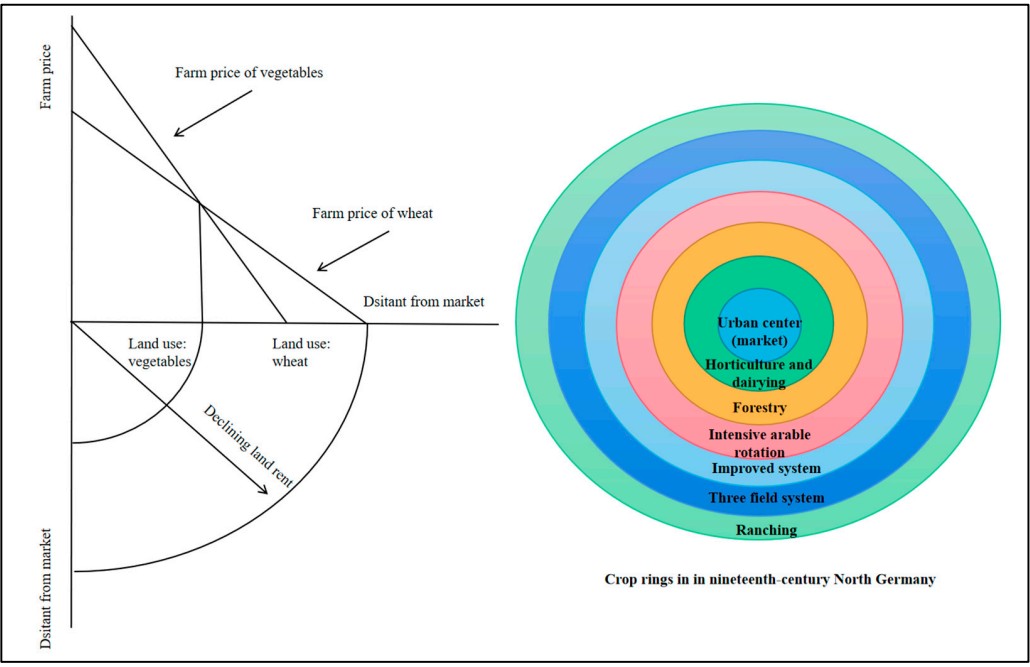

**Figure 1.** Schematic diagram of Von Thünen Model.

The core of Thünen's theory is a question of the comparative advantages of intensive and extensive agricultural enterprises. Intensive and extensive are by nature relative and not absolute conceptions between different crops. Intensive business means a business that uses more capital and labour per unit area [22]. There is an inverse relationship between intensity and distance. Intensity depends on price which depends on transport cost [4]. Garrison and Marble [23] maximized rent as a function of intensity and showed that, given a single production function, the intensity did decrease with distance from the market. Intensity decreases at a decreasing rate as distance to the market increases when there are diminishing marginal returns to intensity [24].

The crop selection mechanism is that crops with higher rents are grown in the vicinity of markets [10]. It is usually the case that sites close to the market are used for products that show the greatest cost increases away from the market. Thünen calculated economic rent for several land uses at several locations, there is a series of concentric rings around the market. In nineteenth-century North Germany, as shown in Figure 1, there were six rings of the Thünen model, which contained, respectively, horticulture and dairying, forestry, intensive arable rotation, improved system, three field system, and, finally, ranching. Beyond the margin of cultivation is a wilderness inhabited only by hunters [4,21] (p. 157).

## 3. OVOP in China

### 3.1. Policy of OVOP in China

As the first developing country to learn from Japan, China incorporated OVOP into its national agricultural policy in 2006. From 2006 to 2019, "OVOP" was mentioned several times in document No. 1. Before 2010, the main task of OVOP in Document No. 1 of China

focused on increasing farmers' income and agricultural industrialization. After 2015, OVOP appeared again in Document No. 1 together with the new concept of "One county, one industry", indicating a shift of policy goals from increasing farmers' income to improving industrial specialization and regional specialization. Further details are given in Table 1.

**Table 1.** OVOP in Document No.1 of China.

| Year | Content of Document |
|------|---------------------|
| 2006 | Promote "OVOP" to actively develop competitive agricultural products to expand the channels for increasing farmers' income |
| 2007 | Cultivate several specialized villages and towns with distinctive characteristics, diverse types, s and, strong competitiveness |
| 2008 | Support the development of "OVOP" to improve the industrialization of agriculture |
| 2010 | Promote the development of "OVOP" and specialized model villages or towns |
| 2015 | Develop "one village, one product, one township (county) and one industry" |
| 2016 | Develop leisure agriculture and rural tourism and build a charming village with "OVOP" |
| 2017 | Build an upgraded version of "one village, one product" |
| 2018 | Establish competitive areas with characteristic agricultural products, and create a new pattern of the development of "OVOP" and "One county, one industry" |
| 2019 | Develop rural characteristic industries: advocate "OVOP" and "One county, one industry" |

Notes: Excerpt from Document No. 1 of China in 2006, 2007, 2008, 2010, 2015, 2016, 2017, 2018 and 2019.

Since OVOP became a national policy in 2006, OVOP has extended to the whole country, and the number of specialized villages and towns in China increased rapidly. By the end of 2020, there were more than 26000 specialized villages and towns in China, accounting for 4.7% of the total 2; meanwhile, 0.8% of the farmers in China have been covered by the "OVOP" movement, which was a population of about 4 million.

*3.2. Crop Varieties of OVOP*

From 2011 to 2020, 3274 villages or towns have been chosen as models nationwide by the Ministry of Agriculture and Rural Affairs of China (MARAC), of which 91 towns had an annual output value of more than 1 billion yuan (157 million dollars or so) and 136 villages had an annual output value of more than 100 million yuan (15.7 million dollars or so).

It is difficult to observe the geographical distribution of different crops due to data availability, but the 3274 specialized model villages or towns (SMTVs) chosen by MARAC provide a unique perspective on agricultural location selection and crop variety selection by analyzing their dominant products. We categorize the dominant products of 3274 SMTVs into six types of crops, including vegetables, fruits, beans and oils, beverages, grains, and livestock, which accounted for about 85% of the total crops. As shown in Figure 2, fruits accounted for 33.08%, vegetables accounted for 27.52%, beverages accounted for 10.87%, grains accounted for 4.58%, livestock accounted for 6.11%, beans and oils accounted for 3.33%, and the rest accounted for 14.51%.

There are different crop mixes from a regional perspective. In eastern China, fruits accounted for 36.05%, vegetables accounted for 29.75%, beans and oils accounted for 3.20%, beverages accounted for 9.01%, grains accounted for 2.33%, livestock accounted for 3.97%, and the rest accounted for 15.70%; In central China, fruits accounted for 25.13%, vegetables accounted for 31.62%, beans and oils accounted for 3.50%, beverages accounted for 12.67%, grains accounted for 6.49%, livestock accounted for 3.81%, and the rest accounted for 16.79%; In western China, fruits accounted for 36.74%, vegetables accounted for 22.58%, beans and oils accounted for 3.30%, beverages accounted for 11.01%, grains accounted for 4.96%, livestock accounted for 9.60%, and the rest accounted for 11.80%.

Vegetables and fruits have always been the two most important crop types. Although the situation in central China is different (vegetables account for the highest proportion), we can see in Figure 2 that the total proportion of vegetables and fruits in the three regions is about 60%. In addition, beverages have been the third highest proportion of products, about 10%, and beans and oils accounted for 3.50%. Finally, the proportion of grains, beans

and oils, and livestock are lower, but the proportion of livestock in the western region is significantly higher.

| | Total | East | Center | West |
|---|---|---|---|---|
| ■ Fruits | 33.08% | 36.05% | 25.13% | 36.74% |
| ■ Vegetables | 27.52% | 29.75% | 31.62% | 22.58% |
| ■ Beans & Oils | 10.87% | 9.01% | 12.67% | 11.01% |
| ■ Beverages | 3.33% | 3.20% | 3.50% | 3.30% |
| ■ Grains | 4.58% | 2.33% | 6.49% | 4.96% |
| ■ Livestock | 6.11% | 3.97% | 3.81% | 9.60% |
| ■ Others | 14.51% | 15.70% | 16.79% | 11.80% |

**Figure 2.** The proportion of dominant products of SMTVs. Notes: Sources from the website of the Ministry of Agriculture and Rural Affairs of China (MARAC); http://www.moa.gov.cn/xw/bmdt/202012/t20201201_6357398.htm (accessed on 1 December 2020).

## 4. Crop Rings Based on Crop Intensity

### 4.1. Crop Location and Crop Intensity

To test the relationship between distance and crop intensity, the first step is to identify the central city. To simplify the analysis, two levels of cities are considered in this paper, they are the smallest county-level cities and the provincial capital city near villages and towns. We used Baidu Maps (like Google Maps) to measure the distances from all SMTVs to county-level cities and provincial capital cities. As shown in Figure 3, the average distances of six dominant crops to county-level cities and metropolises are calculated, vegetables and fruits are the types of crops closest to the cities, followed by beans and oils, beverages, and grains, and livestock are farther away from cities.

In the present work, the method for calculating intensity is based on estimates of different production inputs and activities, which are then converted into a common metric [25]. As shown in Table 2, the output value, total cost, net profits, and cost-profit ratio (Net profit/Total cost) of the six crops were calculated by using the cost and benefit data of agricultural products. The cost and benefit data of major agricultural products are provided by The Compilation of cost-benefit of China's agricultural products, which is published by The National Development and Reform Commission (NDRC) of China. The cost-benefit analysis found that the mean yield, total cost, net profit, and cost-profit ratio of vegetables and fruits are significantly higher than those of other crops, which is the reason why vegetables and fruits accounted for the largest proportion. Beans, oils, and beverages are close in terms of crop intensity. Compared with beans and oils, beverage crops represented by tea have a longer distance, lower cost of land, and higher profits. Due to the national policy of food security and restrictions on farmland use, grain still accounts for about 5%, even if the profit is negative. It was difficult to compare livestock with the other five crops due to differences in units (Unit of livestock: per head, not per Mu). The intensity of livestock, however, is certainly less than those of vegetables, fruits, and beverages by comparing cost-profit ratios.

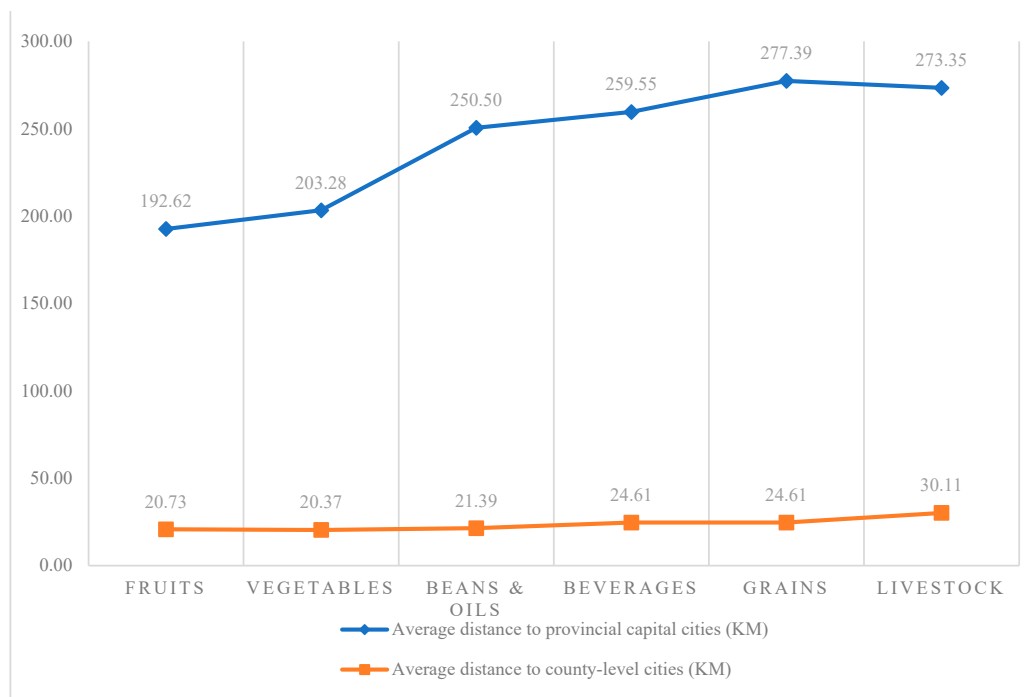

**Figure 3.** Crop varieties and average distance to cities. Notes: The distance shown is a straight line distance, calculated by Baidu Maps.

**Table 2.** Cost-benefit Analysis of Crop intensity.

| Index | Unit | Vegetables | Fruits | Beverages | Beans and Oils | Grains | Livestock |
|---|---|---|---|---|---|---|---|
| Output value | Yuan/Mu | 8225.46 | 5722.51 | 2125.66 | 1262.20 | 1078.40 | / |
| Total cost | Yuan/Mu | 5536.98 | 4064.24 | 1428.04 | 1169.70 | 1108.90 | / |
| Net profit | Yuan/Mu | 2688.48 | 1659.55 | 697.62 | 92.5 | −30.5 | / |
| Cost-profit ratio (Net profit/ Total cost) | % | 48.56 | 46.13 | 48.85 | 7.9 | −2.8 | 19.5 |
| Cost of land | Yuan/Mu | 386.01 | 250.32 | 95.29 | 184.1 | 233.3 | / |
| Numbers of SMTVs | / | 1083 | 901 | 356 | 109 | 150 | 200 |
| Rates of SMTVs | % | 33.10% | 27.54% | 10.88% | 3.33% | 4.58% | 6.11% |

Notes: Sources from The Compilation of cost-benefit of China's agricultural products (2019), Fruits are represented by apple and tangerine; Beans and oils are represented by peanut and rapeseed: Beverages are represented by data of green tea in 2007 because green tea is not collected in statistics after 2007; Grains are the mean value of rice, wheat, and corn; livestock is represented by pig and goat, the output value, total cost, net profit and the cost of land are not shown as the unit of measurement for livestock is different from other crops (Unit of livestock: per head).

The source of the difference in crop intensity is the land rent (or location rent), which is defined as net income by Thünen. Net revenue, which varies by crop and as a function of distance, is what yields the zonation of crop areas as concentric bands that radiate from the market centre in widening circles [10]. Based on the above analysis, three types of crops can be distinguished, namely high-density crops, medium-density crops, and low-density crops. High-density crops, represented by vegetables with net revenue of 2688.48 Yuan/Mu and fruits with net revenue of 1659.55 Yuan/Mu, account for more than 60% of crop types. High-density crops with the characteristics of high land cost, high input intensity, and high profit are closest to cities. The land cost, input intensity, and profit of medium-density crops are moderate, and the average distance from cities is higher than that of high-density crops. Medium-density crops are represented by beans and oils with net revenue of 92.50 Yuan/Mu and beverages with net revenue of 697.62 Yuan/Mu, accounting for about 15% of crop types. Low-density crops, represented by grains with net

revenue of −30.5 Yuan/Mu and livestock, have low profits and are farthest from cities, and their proportions are all below 5%.

### 4.2. Crop Rings of the Thünen Model

#### 4.2.1. Rings around County-Level Cities

In the scenario of The Isolated Country, economic activities are distributed according to the pattern of concentric rings, each of them being specialized in one crop [26] (p. 145). In reality, the crop structure of each region is a mixture of multiple crops. To observe the differences between locations, five rings were divided according to the distance from the city, and the proportion of crops in each ring was the focus of our attention.

Centred on county-level cities, the first ring is the area within five kilometres of the county-level city, the second ring is the area of 5 to 10 km, the third ring is the area of 10 to 20 km, and the fourth ring is the area of 20 to 30 km, the fifth outermost ring is the area 30 km away. As shown in Figure 4, the structural composition of the innermost ring is 41.23% for fruits, 33.77% for vegetables,5.26% for beverages, 2.19% for beans and oils, 2.63% for grains, and 1.75% for livestock, whereas the outermost ring has a crop composition of 25.57% are fruits, 22.07% are vegetables, 16.13% are beverages, 4.11% are beans and oils, 6.39% are grains, and 7.76% are livestock.

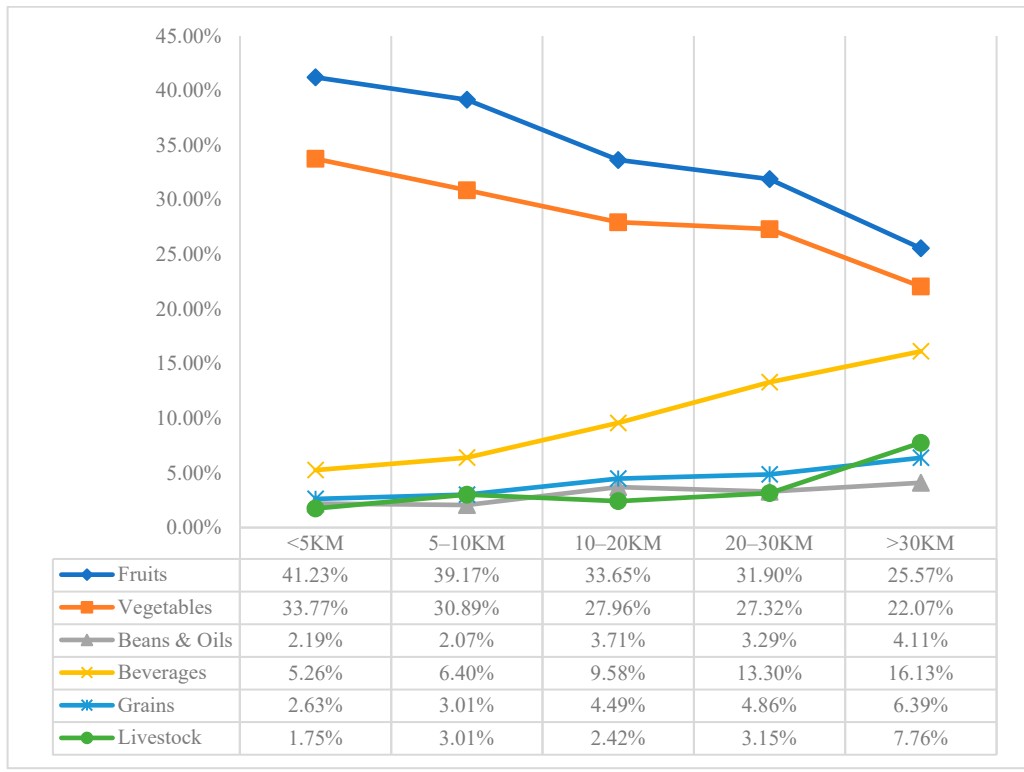

| | <5KM | 5–10KM | 10–20KM | 20–30KM | >30KM |
|---|---|---|---|---|---|
| Fruits | 41.23% | 39.17% | 33.65% | 31.90% | 25.57% |
| Vegetables | 33.77% | 30.89% | 27.96% | 27.32% | 22.07% |
| Beans & Oils | 2.19% | 2.07% | 3.71% | 3.29% | 4.11% |
| Beverages | 5.26% | 6.40% | 9.58% | 13.30% | 16.13% |
| Grains | 2.63% | 3.01% | 4.49% | 4.86% | 6.39% |
| Livestock | 1.75% | 3.01% | 2.42% | 3.15% | 7.76% |

**Figure 4.** Crop rings around county-level cities. Notes: Sources from the website of the Ministry of Agriculture and Rural Affairs of China (MARAC); http://www.moa.gov.cn/xw/bmdt/202012/t20201201_6357398.htm (accessed on 1 December 2020).

From the innermost ring to the outermost ring, although high-density crops have always dominated, the proportion of high-density crops has been decreasing, and the medium-density and low-density crops have shown an upward trend. Fruits and vegetables were the crops with the highest proportions for all rings, but both showed a decreasing trend with increasing distance from county-level cities. From the first ring to the fifth ring, the proportion of fruits declined by 15.66%, and the proportion of vegetables dropped by 11.70%. Instead, we can see a trend of rising proportions of beans and oils, beverages, grains, and livestock with increasing distance. Among them, from the innermost ring to

the outermost ring, the proportion of beverages increased by 3 times, the proportion of grains increased by 3 times, the proportion of livestock increased by more than 2 times, and the proportion of beans and oils also increased by nearly 2 times.

### 4.2.2. Rings around Provincial Capital Cities

Around provincial capital cities, five rings are also divided to observe changes in crop structures. The first ring is the area within 50 km of the county-level city, the second ring is the area of 50 to 100 km, the third ring is the area of 100 to 150 km, and the fourth ring is the area of 150 to 200 km, the outermost ring is the area 200 km away. In Figures 5 and 6, the structural composition of the innermost ring is 34.33% for fruits, 33.27% for vegetables, 8.27% for beverages, 2.29% for beans and oils, 4.05% for grains, and 2.46% for livestock, whereas the outermost ring has a crop composition of 35.50% are fruits, 24.41% are vegetables, 12.52% are beverages, 3.49% are beans and oils, 4.44% are grains, and 5.23% are livestock.

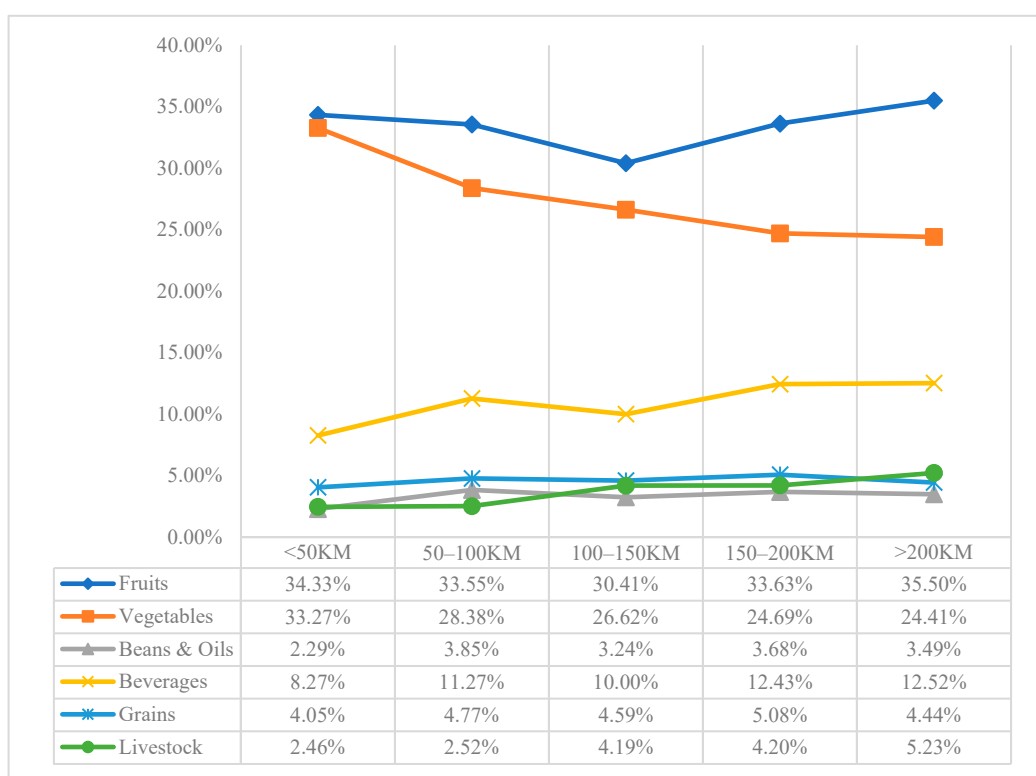

**Figure 5.** Crop rings around provincial capital cities. Notes: Sources from the website of the Ministry of Agriculture and Rural Affairs of China (MARAC); http://www.moa.gov.cn/xw/bmdt/202012/t20201201_6357398.htm (accessed on 1 December 2020).

Overall, similarly, from the first ring to the fifth ring, the total proportion of high-density crops decreased, and the total proportion of medium-density and low-density crops increased. Fruits and vegetables are the crops with the highest proportion of all rings, and the total share of the two crops is decreasing. The difference is that vegetables show a downward trend with increasing distance, whereas the proportion of fruits increases in the fourth and fifth rings. At the same time, the proportions of beverages and livestock have a slightly increasing trend with increasing distance, while beans and oils, and grains have been relatively stable.

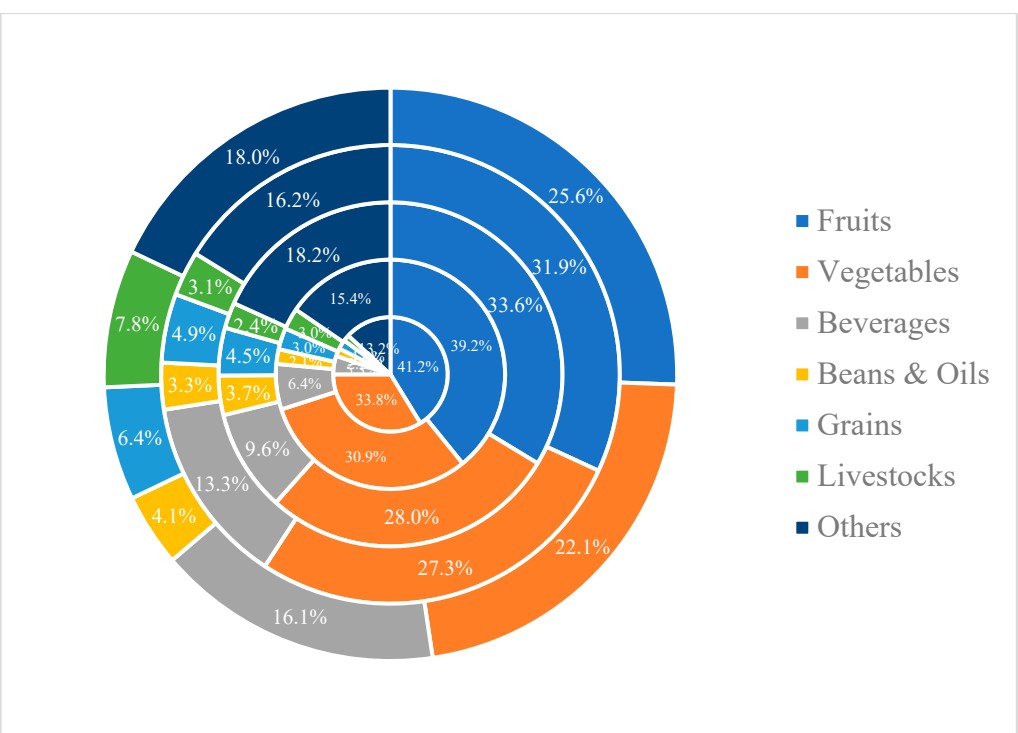

**Figure 6.** Concentric rings of crops around provincial capital cities.

*4.3. Regional Heterogeneity of Crop Rings*

> "*A basic principle in economics is a particular location may specialize in a particular activity for two broadly defined reasons. First, the location might have some underlying characteristic that gives it a natural advantage in the activity. Second, some type of scale economy may be attained by concentrating production at the location. When we observe specialization, we can ask about the roles these two factors play*" [27] (p. 1).

According to the Von Thünen model, the urban marketplace transmitted development impulses to an expanding hinterland [15]. In the unprecedented process of urbanization in China, the growth rate of urbanization is twice that of the world in the same period [28]. Urban growth and hinterland were coupled, with sprawl "pushing" agriculture ever farther afield [10]. Crop rings around county-level cities in different regions were used to discuss heterogeneity. Heterogeneity analysis found that the proportion of high-density crops was higher in eastern China; there is a balanced structure of crops in central and western China. This indicates that the influence of urban forces in eastern regions with higher urbanization rates and denser populations extended to more distant regions.

Specifically, vegetables and fruits in eastern China occupy an absolute dominant position. As distance increases, the share of fruits and vegetables decreases, part of the share is replaced by beverage crops, and other crops are almost absent. Farmers in eastern China are more concentrated on high-density and high-margin crops due to denser populations, larger cities, and higher labour and land costs. In central China, beverage crops have gradually increased to be close to vegetables and fruits, and the share of cereals has also increased with increasing distance. Because central China is the main tea-producing and grain-producing region of China, the share of beverage crops and cereals is higher than those in other regions; western China is characterized by a balanced crop structure. With the increase in distance, the proportion of grains, beverages, livestock, as well as beans and oils all increased. Low population density, large areas of arable land, and more grassland are the characteristics the western China, which has led to the formation of a development model that utilizes the comparative advantages of labour and natural resources in the western region, and its crop structure is relatively balanced (See Table 3).

**Table 3.** Regional heterogeneities of crop rings around county-level cities.

| Region | Varieties | <5KM | 5–10KM | 10–20KM | 20–30KM | >30KM |
|--------|-----------|------|--------|---------|---------|-------|
| East | Fruits | 48.15% | 45.25% | 33.65% | 33.47% | 30.95% |
| | Vegetables | 29.63% | 28.49% | 31.06% | 33.06% | 20.63% |
| | Beans and Oils | 5.56% | 0.00% | 4.94% | 1.61% | 3.97% |
| | Beverages | 5.56% | 3.91% | 7.53% | 11.69% | 17.46% |
| | Grains | 0.00% | 2.79% | 2.35% | 2.82% | 1.59% |
| | Livestock | 0.00% | 1.12% | 1.65% | 1.61% | 1.59% |
| Center | Fruits | 48.61% | 37.82% | 30.29% | 27.98% | 26.67% |
| | Vegetables | 23.61% | 29.49% | 22.35% | 30.05% | 22.38% |
| | Beans and Oils | 2.78% | 5.77% | 3.82% | 3.11% | 1.90% |
| | Beverages | 5.56% | 3.85% | 10.88% | 13.99% | 23.33% |
| | Grains | 4.17% | 3.21% | 7.35% | 7.77% | 7.14% |
| | Livestock | 0.00% | 1.92% | 2.06% | 1.04% | 2.38% |
| West | Fruits | 48.91% | 37.34% | 43.07% | 29.77% | 24.09% |
| | Vegetables | 28.26% | 32.91% | 23.80% | 23.26% | 18.81% |
| | Beans and Oils | 0.00% | 1.27% | 2.71% | 5.58% | 5.94% |
| | Beverages | 5.43% | 10.76% | 9.64% | 10.70% | 11.22% |
| | Grains | 3.26% | 3.80% | 4.52% | 5.12% | 8.25% |
| | Livestock | 3.26% | 6.33% | 4.22% | 6.98% | 14.19% |

*4.4. Cross-Tabulation Analysis of Metropolitan and County-Level Cities*

To consider the impact of both metropolitan and county-level cities on crop types, four types of areas have been identified according to the distance to the two types of cities. Randolph [29] categorizes the peri-urban interface as a 10 km zone surrounding the cities in Australia. US planners define peri-urban as an area that is located within a 60–70mile radius of all metropolitan areas [30]. Webster [31] estimates a 100–300 km zone of peri-urban in East Asia and China. Hence, the proximity areas of county-level cities and metropolitan cities were set to be within 10 km and 100 km, respectively. The first type that we defined is the dual peri-urban area, which is an area adjacent to a metropolitan city (within 100 km) and also adjacent to a county-level city (within 10 km); the second type is the peri-metropolitan area, which is an area adjacent to a metropolitan city (within 100 km) but far away from a county-level city (more than 10 km); the third type is the peri-county area, which is an area far metropolitan area (more than 100 km) but adjacent to a county-level city (within 10 km); and the fourth type is the suburban and rural area, which is an area far from a metropolitan area (more than 100 km) and far from a county-level city (more than 10 km).

Cross-tabulation analysis reveals that crop varieties are jointly influenced by cities at different levels. The proximity to multiple cities further increases the proportion of high-density crops. As shown in Table 4, the dual peri-urban area adjacent to the two types of markets has the highest proportion of high-intensity crops of vegetables and fruits, the suburban and rural area has the lowest proportion of high-intensity crops, and the proportion of high-intensity crops in the peri-metropolitan area and the peri-county area are in the middle. Specifically, the peri-county region preferred fruits, and the peri-metropolitan region preferred vegetables. In terms of medium-intensity crops and low-intensity crops, Beans and Oils were evenly distributed in four areas; Beverages, Grains, and livestock were distributed in the same pattern, with the lowest proportion in the dual peri-urban area, followed by the proportion in the peri-county area, the third proportion in peri-metropolitan area, the highest proportion in the suburban and rural area.

**Table 4.** Cross-tabulation analysis.

| Crop Variety | Dual Peri-Urban (<10 & <100) | | Peri-Metropolitan (>10 & <100) | | Peri-County (<10 & >100) | | Suburban and Rural (>10 & >100) | |
|---|---|---|---|---|---|---|---|---|
| | Numbers | Ratios | Numbers | Ratios | Numbers | Ratios | Numbers | Ratios |
| Fruits | 80 | 37.74% | 184 | 31.35% | 222 | 40.59% | 597 | 30.96% |
| Vegetables | 74 | 34.91% | 205 | 34.92% | 167 | 30.53% | 455 | 23.60% |
| Beans and Oils | 7 | 3.30% | 17 | 2.90% | 9 | 1.65% | 76 | 3.94% |
| Beverages | 10 | 4.72% | 43 | 7.33% | 36 | 6.58% | 267 | 13.85% |
| Grains | 0 | 0.00% | 30 | 5.11% | 18 | 3.29% | 98 | 5.08% |
| Livestock | 9 | 4.25% | 31 | 5.28% | 28 | 5.12% | 135 | 7.00% |
| Total | 212 | 84.92% | 587 | 86.89% | 547 | 87.76% | 1928 | 84.43% |

## 5. Conclusions

Driven by the policy of one village and one product, crop structure has changed from diversification to highly specialization to some extent. Market-oriented farmers tend to use local resources as a comparative advantage by choosing competitive crop varieties. The locational layout of agriculture in China is bound to revolve around cities, which is the huge consumer market. As China continues to urbanize, more rural residents migrate to cities, and the share of food consumption in cities will maintain further growth. Based on the metropolitan ring centred on the provincial capital city and the small-city ring centred on the county-level city, we observed the structural changes of crops at different distances from the city and found the following characteristics: (1) The phenomenon of crop intensity decreasing with distance described by the Thünen model still exists. Among the multiple crop combinations around the city, with the increase in distance from the city, the high-intensity crops of fruits and vegetables continued to decline, while the proportion of medium-density crops and low-density crops increased. Therefore, the total agricultural intensity decreases with the increase of distance; (2) The crop theory of the Thünen model still applies to OVOP in China, city-centered crop rings still exist, especially obvious around county-level cities. As the crop ring expands outward, the proportion of high-density crops (vegetables and fruits) decreases, while the proportions of medium-density (beans and oils, and beverages) and low-density (grains and livestock) farming activities increase; (3) Under the trend of agricultural commercialization and specialization, high-intensity crops are the best choice for the specialized villages and towns. In all rings, vegetables and fruits dominate, followed by pulses and oil crops, and beverage crops, while proportions of cereals and livestock products are smaller; (4) The analysis of regional heterogeneity shows that eastern China has a greater range of urban influence and its crop density is higher at the same market distance; in central China and western China, with lower urbanization rates and less dense populations, has a progressively more balanced crop structure with increasing distance due to limited urban influence. Cross-tabulation analysis showed that crop densities decreased sequentially in dual peri-urban, peri-county, peri-metropolitan, and suburban and rural areas. An area adjacent to multiple municipalities tends to have a higher crop density and a higher proportion of high-density crops.

One of the key issues facing China's agriculture is how to achieve specialization. China's central government is promoting the construction of special agricultural product advantage zones and proposing "one county, one industry", which is based on "one village, one product", to encourage counties to focus on advantageous industries to achieve agricultural specialization. Therefore, the suggestions are as follows: First, the cultivated land around the city should be distributed according to the urban demand; Secondly, in addition to endowment and natural conditions, the selection of crop varieties must take into account the distance from the market (metropolis, neighbouring cities); At the same time, the principle that crop density decreases with market distance still has practical significance; Finally, the agglomeration effect of urban agglomeration has a greater impact on crop distribution, and planning of farmland needs to consider the impact of surrounding cities.

It must be emphasised that this paper shows that the basic principles of Thünen's theory are still extremely valuable, by analyzing the distribution and layout of different crops around cities (provincial capital cities, county-level cities) in China. Further research is needed on how to select crop varieties and organize agricultural production around the city in a more scientific way.

**Author Contributions:** Conceptualization, H.H. and Z.Y.; methodology, Z.Y.; software, Z.Y.; validation, H.H., Z.Y. and K.Z.; formal analysis, Z.Y.; writing—original draft preparation, H.H., Z.Y.; writing—review and editing, H.H., Z.Y.; visualization, K.Z.; funding acquisition, H.H. All authors have read and agreed to the published version of the manuscript.

**Funding:** This research was funded by the National Natural Science Foundation of China, grant number 72073119.

**Data Availability Statement:** Not applicable.

**Conflicts of Interest:** The authors declare no conflict of interest.

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
