# Peer review of "Agricultural Location and Crop Choices in China: A Revisitation on Von Thünen Model"

_land, doi:10.3390/land11111885_

Round 1
Reviewer 1 Report
-The paper claims that Agricultural Location and Crop Selection still follow the Thunen Model. In a way, this is very relevant to explore and Authors have done so nicely with a strong scientific rigour. However, the paper lacks to critically discuss the context of socio-economic transition in China in a broader sense that must be affecting the predictive model like this. For example, globally agriculture production is being impacted from out-migration and urbanisation. Agriculture labor are the biggest challenges. Analysis /discussion on the impact of these factors in Thunen model, would have strengthen the article.
-A clear and elaborative introduction of Thunen Model will help to understand the context and relevance of the article in advance. Authors have directly jumped on the views of scholars about the model in introduction, although there is a detailed section of this Model later, readers will be confused first. Likewise, need to explain what “economic Geography approach” means ?
Author Response
Dear reviewer,
I would like to thank you for your efforts in reviewing the manuscript titled “Agricultural location and crop choices in China: A revisitation on Von Thünen Model” and providing many helpful comments and suggestions, which are all invaluable in the revision and improvement of this paper, as well as in guiding my research in the future.
I have studied your comments point by point and revised the manuscript accordingly.
Point 1
The paper claims that Agricultural Location and Crop Selection still follow the Thunen Model. In a way, this is very relevant to explore and Authors have done so nicely with a strong scientific rigour. However, the paper lacks to critically discuss the context of socio-economic transition in China in a broader sense that must be affecting the predictive model like this. For example, globally agriculture production is being impacted from out-migration and urbanisation. Agriculture labor are the biggest challenges. Analysis /discussion on the impact of these factors in Thunen model, would have strengthen the article.
Response to Point 1
Thank you for your comment. Following your suggestion, we add some words in conclusions. The added content is “As China continues to urbanize and more rural residents migrate to cities, the share of food consumption in cities will maintain further growth.”, which is in line 363, on page 12.
Point 2
A clear and elaborative introduction of Thunen Model will help to understand the context and relevance of the article in advance. Authors have directly jumped on the views of scholars about the model in introduction, although there is a detailed section of this Model later, readers will be confused first. Likewise, need to explain what “economic Geography approach” means ?
Response to Point 2
Thank you for your comment. We have added an brief introduction of the model in the abstract, the added content is “Von Thünen Model, which is fronted by 19th-century German economist, outlines a rural landscape of commercial farmers growing agricultural products for local markets, while proposing basic patterns and principles of land use in agriculture”. The research problems of economic geography are how the spatial dimension of the economy is organized, that is, where production takes place, where consumption takes place, through which the exchange of rare goods takes place. This article mainly answers the question: What crops should be produced in a specific region? Regional Approach, Systematic or Commodity Approach, Activity Approach and Principles Approach in economic geography are all used.
Reviewer 2 Report
1. The paper mainly starts from the basic principles of Von Thünen Model, which is conducive to the scientific utilization of land space and the effective optimization of agricultural production, and at the same time, continuously promotes the development of agriculture in the future. The topic selection has certain practical significance, but the analysis of the research significance of the paper is relatively Weak.
1. The thesis and methods are justified. Using data from China’s OVOP (“One Village, One Product”), the agricultural location and crop selection around two levels of cities (provincial capital cities, and county-level cities) are analyzed.It is proved that China's agricultural location and crop selection conform to the basic principle of Von Thünen Model, and the conclusion of this paper is highly convincing. However, there are many descriptions of the present situation in this paper. It is suggested to supplement how to act on China's next agricultural development.
1. The structure of the paper is relatively complete. It analyzes the location and crop selection of Chinese crops from five aspects: the first part of the introduction, the second part of the introduction of Von Thünen Model, the third part of combing China's OVOP (“One Village, One Product”), the fourth part of analyzing the location and intensity of crops, and the fifth part of the conclusion. However, it is suggested to introduce a new part to put forward policy suggestions for agricultural development.
4. The fourth part of the paper analyzes the location and intensity of crops from four aspects. At the same time, a large number of charts are used to support arguments, and the data are detailed. However, considering the reliability of data sources, it is suggested to add data sources below the charts, such as Figure 1 and Figure 2.
Author Response
Dear reviewer,
I would like to thank you for your efforts in reviewing the manuscript titled “Agricultural location and crop choices in China: A revisitation on Von Thünen Model” and providing many helpful comments and suggestions, which are all invaluable in the revision and improvement of this paper, as well as in guiding my research in the future.
I have studied your comments point by point and revised the manuscript accordingly.
Point 1
The paper mainly starts from the basic principles of Von Thünen Model, which is conducive to the scientific utilization of land space and the effective optimization of agricultural production, and at the same time, continuously promotes the development of agriculture in the future. The topic selection has certain practical significance, but the analysis of the research significance of the paper is relatively Weak.
Response to Point 1
Thank you for your comment. Following your suggestion, we further strengthened the innovation of the article. The added content is “ (3) Compared with the previous studies that only focus on the crop ring of a single city, the cross impacts from different levels of cities are distinguished and studied.”, which is in line 77, on page 2.
Point 2
The thesis and methods are justified. Using data from China’s OVOP (“One Village, One Product”), the agricultural location and crop selection around two levels of cities (provincial capital cities, and county-level cities) are analyzed.It is proved that China's agricultural location and crop selection conform to the basic principle of Von Thünen Model, and the conclusion of this paper is highly convincing. However, there are many descriptions of the present situation in this paper. It is suggested to supplement how to act on China's next agricultural development.
The structure of the paper is relatively complete. It analyzes the location and crop selection of Chinese crops from five aspects: the first part of the introduction, the second part of the introduction of Von Thünen Model, the third part of combing China's OVOP (“One Village, One Product”), the fourth part of analyzing the location and intensity of crops, and the fifth part of the conclusion. However, it is suggested to introduce a new part to put forward policy suggestions for agricultural development.
Response to Point 2
Thank you for your comment. Following your suggestion, we have added a paragraph of policy recommendations in the article. The added content is “One of the key issues facing China’s agriculture is how to achieve specialization. China's central government is promoting the construction of special agricultural product advantage zones and proposing "one county, one industry", which is based on "one village, one product", to encourage counties to focus on advantageous industries to achieve agricultural specialization. Therefore, the suggestions are as follows: First, the cultivated land around the city should be distributed according to the urban demand; Secondly, in addition to endowment and natural conditions, the selection of crop varieties must take into account the distance from market (metropolis, neighboring cities); At the same time, the principle that crop density decreases with market distance still has practical significance; Finally, the agglomeration effect of urban agglomeration has a greater impact on crop distribution, and planning of farmland needs to consider the impact of surrounding cities.”, which is in line 389, on page 12.
Point 3
The fourth part of the paper analyzes the location and intensity of crops from four aspects. At the same time, a large number of charts are used to support arguments, and the data are detailed. However, considering the reliability of data sources, it is suggested to add data sources below the charts, such as Figure 1 and Figure 2.
Response to Point 3
Thank you for your comment. Thank you for your comment. Following your suggestion, we add data sources below the charts. The added content is “Notes: Sources from website of the Ministry of Agriculture and Rural Affairs of China (MARAC) ; http://www.moa.gov.cn/.”, which is on page 5 (below the Figure 2), page 7 (below the Figure 4), and page 9 (below the Figure 5). We also added “Notes: The distance shown is a straight line distance, calculated by Baidu Maps.” on page 6 (below the Figure 3).
Reviewer 3 Report
1. Please explain briefly about the Von Thünen Model in the abstract. 2. The introduction is written in a manner that the present article seems to be a review of the Von Thünen Model. Please explain the model in the perspective of your study. Focus on highlighting the aim and objectives of your study in the introduction. 3. Provided a schematic diagram of Von Thünen Model in section 2. 4. Provide the full form of SMTVs in the table 1. 5. The discussion needs to be more centered on OVOP and its impact on crop structure. 6. Section 4.2.2 requires a diagrammatic representation with concentric rings representing the changes in crop structure.Author Response
Dear reviewer,
I would like to thank you for your efforts in reviewing the manuscript titled “Agricultural location and crop choices in China: A revisitation on Von Thünen Model” and providing many helpful comments and suggestions, which are all invaluable in the revision and improvement of this paper, as well as in guiding my research in the future.
I have studied your comments point by point and revised the manuscript accordingly.
Point 1
Please explain briefly about the Von Thünen Model in the abstract.
Response to Point 1
Thank you for your comment. We have added an explanation of the model in the abstract, The added content is “Von Thünen Model, which is fronted by 19th-century German economist, outlines a rural landscape of commercial farmers growing agricultural products for local markets, while proposing basic patterns and principles of land use in agriculture”, which is on page 1.
Point 2
The introduction is written in a manner that the present article seems to be a review of the Von Thünen Model. Please explain the model in the perspective of your study. Focus on highlighting the aim and objectives of your study in the introduction.
Response to Point 2
Thank you for your comment. Follow you advice, after review of the Von Thünen Model, we directly emphasized the purpose of the study. The content is “Associated with the change of evolving economic and resource circumstances, as improved transportation conditions have led to a decrease in transportation costs, whether the Thünen model still has explanatory power is a concern of recent research. If so, does the Thünen model still apply, if it is yes, and to what extent it is applied to the explanation of the distribution of crop rings? Further research on the applicability of the Thünen model is needed for a better understanding of agricultural patterns in developing countries, which helps to optimize their land use and agricultural productivity”.
Point 3
Provided a schematic diagram of Von Thünen Model in section 2.
Response to Point 3
Thank you for your comment. We have added a schematic diagram of Von Thünen Model in section 2, which is figure 1.
Point 4
Provide the full form of SMTVs in the table 1.
Response to Point 4
Thank you for your comment. Following your suggestion, the full form of SMTVs in Table 1 has been provided.
Point 5
The discussion needs to be more centered on OVOP and its impact on crop structure.
Response to Point 5
Thank you for your comment. As you suggested above, more attention has been given to the crop structure in the development of OVOP. The content is “As the crop ring expands outward, the proportion of high-density crops (vegetables and fruits) decreases, while the proportions of medium-density (beans and oils, and beverages) and low-density (grains and livestock) farming activities increase; (3) Under the trend of agricultural commercialization and specialization, high-intensity crops are the best choice for the specialized villages and towns. In all rings, vegetables and fruits dominate, followed by pulses and oil crops, and beverage crops, while proportions of cereals and livestock products are smaller;(4) The analysis of regional heterogeneity shows that eastern China has a greater range of urban influence and its crop density is higher at the same market distance; in central China and western China, with lower urbanization rates and less dense populations, has a progressively more balanced crop structure with increasing distance due to limited urban influence. Cross-tabulation analysis showed that crop densities decreased sequentially in dual peri-urban, peri-county, peri-metropolitan, and suburban and rural areas. An area adjacent to multiple municipalities tends to have a higher crop density and a higher proportion of high-density crops”.
Point 6
Section 4.2.2 requires a diagrammatic representation with concentric rings representing the changes in crop structure.
Response to Point 6
Thank you for your comment. We have added a diagram with form of concentric rings in section 4.2.2, which is figure 6 on page 9.
Reviewer 4 Report
I have gone through the manuscript entitled "Agricultural location and crop choices in China: A revisitation on Von Thünen Model" and found it interesting as well as satisfying at the current level. In my opinion it can be published as such.
Author Response
Dear reviewer,
I would like to thank you for your efforts in reviewing the manuscript entitled "Agricultural location and crop choices in China: A revisitation on Von Thünen Model". We are encouraged by your affirmation of our work and wish you a pleasant work.
Round 2
Reviewer 2 Report
1. Using China's OVOP ("One Village, One Product") data, starting from the basic principles of the Von Thünen Model, this paper analyzes the agricultural location and crop selection around two levels of cities (provincial capital cities, and county-level cities), proving that China's agricultural location and crop selection can learn from the basic principles of the Von Thünen Model, which is conducive to continuously promoting the development of China's agriculture in the future. The topic has certain practical significance.
2. The structure of the paper is relatively complete, including the introduction of the first part, the introduction of Von Thünen Model in the second part, the carding of China's OVOP ("one village, one product") in the third part, the analysis of crop growth rings based on crop intensity in the fourth part, and the conclusion in the fifth part. The paper is relatively complete in structure and clear in research ideas.
3. In the analysis and result presentation part of the fourth chapter, whether the mathematical logic should be combed more clearly to make the analysis and result presentation clearer and increase the readability of readers.
Reviewer 3 Report
The authors have thoroughly revised the manuscript and addressed all the comments satisfactorily. Thus, the manuscript can be accepted in its present form.